# Out-of-Distribution Detection based on In-Distribution Data Patterns Memorization with Modern Hopfield Energy

**Jinsong Zhang [1][*][†], Qiang Fu [2][*][‡], Xu Chen [2], Lun Du [2], Zelin Li [2][†], Gang Wang [1], XiaoGuang Liu [1], Shi Han [2], Dongmei Zhang [2]**

[1] College of Computer Science, Nankai University, Tianjin, China

[2] Microsoft Corporation, Beijing, China

{zhangjs,wgzwp,liuxg}@nbjl.nankai.edu.cn

{qifu,xu.chen,lun.du,v-zelinli,shihan,dongmeiz}@microsoft.com

## Abstract

Out-of-Distribution (OOD) detection is essential for safety-critical applications of deep neural networks. OOD detection is challenging since DNN models may produce very high logits value even for OOD samples. Hence, it is of great difficulty to discriminate OOD data by directly adopting Softmax on output logits as the confidence score. Differently, we detect the OOD sample with Hopfield energy in a store-then-compare paradigm. In more detail, penultimate layer outputs on the training set are considered as the representations of in-distribution (ID) data. Thus they can be transformed into stored patterns that serve as anchors to measure the discrepancy of unseen data for OOD detection. Starting from the energy function defined in Modern Hopfield Network for the discrepancy score calculation, we derive a simplified version **SHE** with theoretical analysis. In SHE, we utilize only one stored pattern to present each class, and these patterns can be obtained by simply averaging the penultimate layer outputs of training samples within this class. SHE has the advantages of hyperparameter-free and high computational efficiency. The evaluations of nine widely-used OOD datasets show the promising performance of such a simple yet effective approach and its superiority over State-of-the-Art models. Code is available at https://github.com/zjs975584714/SHE_ood_detection.

## 1 Introduction

Deep Neural Network (DNN) has yielded remarkable achievements in a broad range of fields in recent years (He et al., 2016; Huang et al., 2017), and is extensively deployed in numerous real-world scenarios (Krizhevsky et al., 2017; Redmon & Farhadi, 2017). One of its powerful capabilities lies in the promising generalization ability from training data to unseen in-distribution (ID) data. However, the finite training data cannot guarantee the completeness of data distribution, so it is inevitable to encounter out-of-distribution (OOD) data. The Softmax-based prediction allows OOD samples to gain high confidence in specific classes, which is unacceptable in practice, especially for safety-related areas. It can lead to erroneous collisions in autonomous driving or irreparably large financial losses. Therefore, OOD detection is critical with respect to AI safety (Amodei et al., 2016).

Existing efforts on OOD detection for DNN can be roughly divided into two categories. The first group of approaches requires designing and retraining new auxiliary networks specifically for OOD detection rather than directly using already trained models (Denouden et al., 2018; DeVries & Taylor, 2018; Yu & Aizawa, 2019; Zhang et al., 2020). The objective should be modified accordingly and OOD samples are sometimes introduced to train the new networks. However, it is almost impossible to exhaust all kinds of OOD samples, and retraining can also be cumbersome. The methods

---

[*]Equal contribution

[†]This work was done during interning at Microsoft.

[‡]Corresponding author.

of the second category elaborate on the confidence design for the network outputs, e.g., the logits, the Softmax probability (Liang et al., 2017; Liu et al., 2020; Sun et al., 2021) or embedding features (Lee et al., 2018; Sehwag et al., 2021; Sun et al., 2022). By these means, there is no need to modify the backbone model and the objective, which motivates us to explore OOD detection in this manner.

In deep learning, the intermediate layer output can be regarded as the representation of input data in the latent space. Further, as shown in Figure. 1 (left), guided by the training process, such representations of ID samples of the same category tend to present some common patterns for prediction accuracy. In contrast, these representations of OOD samples should deviate from such commonality since they are not considered during the training process. Based on this intuition, **OOD detection can be formulated as a store-then-compare process**: representations of ID samples within each category are maintained during the training procedure as stored patterns, and a test pattern will be compared with the store patterns. If there is a noticeable discrepancy, then it can be judged as an OOD sample.

The critical question is ***how to measure the discrepancy between the OOD sample and the stored patterns under this setting?*** To accomplish this goal, we adopt the key idea of a classic memory network, Hopfield Network. The Hopfield Network (binary state) was first introduced in (Hopfield, 1982) and (Hopfield, 1984) proposed continuous state version. Modern Hopfield Network (both continuous and binary) was introduced in (Krotov & Hopfield, 2016), and (Ramsauer et al., 2020) proposed a new energy function for continuous state Hopfield networks and point out the relationship with the transformer.

Hopfield Network targets recovering distorted test patterns so that the recovered patterns are as close to the stored patterns as possible. It achieves this goal by specific update rules that minimize the predefined energy function. The more the recovered pattern resembles the stored pattern, the lower the energy is. Therefore, the energy function serves a vital role as it indicates the gap between the recovered patterns and the stored patterns. For OOD detection tasks, the energy function of the Hopfield Network is well-suited as a desirable measure of the discrepancy between the OOD sample and the stored patterns.

In this paper, we propose a new OOD detection method **HE** with memorization of ID data patterns and the Modern **H**opfield **E**nergy function (Ramsauer et al., 2020). In more detail, the representations of training ID samples are stored as patterns for each category in advance, and OOD samples are detected under the energy function. As the intermediate results are more informative than the highly-compressed final output logits, we preserve the outputs of the penultimate layer (i.e., the input of the final output layer) as representations. Furthermore, to address the challenges of the memory cost of pattern memorization, we derive a **S**implified **H**opfield **E**nergy function-based method **SHE**. In **SHE**, only one pattern is required for each category and there is no hyperparameter to be tuned. Theoretical analysis proves the effectiveness of our design. The remarkable performances on nine widely-used OOD detection datasets on three different networks demonstrate the superiority of our proposed SHE (and HE) over state-of-the-art methods. We summarize the main contributions of our paper as follows:

- We propose a Modern Hopfield Energy-based method HE for out-of-distribution detection. It uses store-then-compare paradigm that compares test samples with pre-stored patterns to measure the discrepancy from in-distribution data according to Hopfield energy.

- We derive a simplified version of HE, named as SHE, which greatly reduces the memory and the computation cost. In addition, SHE is hyperparameter-free. Theoretical analysis is conducted to illustrate the effectiveness of SHE.

- Extensive experiments on nine OOD detection datasets of three prominent computer vision backbone networks indicate both the effectiveness and the efficiency of our designed methods. Experiments on large-scale datasets (e.g., ImageNet-1k) also show the superior of our approach. In-depth analysis and ablation studies are also included to shed light on the mechanism behind it.

## 2   RELATED WORK

**Network Redesign and Retrain.** Given original network architectures are designed for target tasks like classification, a straightforward paradigm of OOD detection is to elaborate on the network ar-

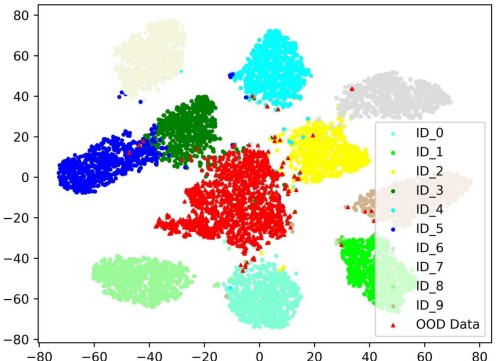 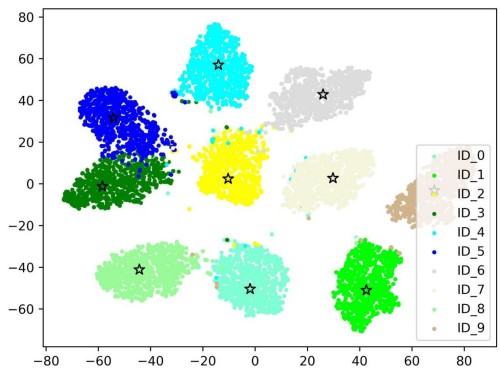

Figure 1: Visualization of the distribution of ID/OOD patterns (left) and ID/Stored patterns (right) by t-SNE (Van der Maaten & Hinton, 2008), the ID/OOD refers to CIFAR10 (Krizhevsky et al., 2009) and LSUN (Yu et al., 2015), respectively. The backbone network is ResNet18 (He et al., 2016). In the left figure, the OOD patterns are away from the ID patterns. In the right figure, each pentagram denotes the stored pattern in SHE corresponding to each category, from the figure we can see the stored pattern can represent the corresponding ID patterns well.

chitecture and the objective, and retrain a new network for OOD detection. (Denouden et al., 2018) used ID data to train an AutoEncoder, and OOD samples were supposed to have high reconstruction errors. (DeVries & Taylor, 2018) expanded the original network with an extra branch to predict a confidence score and obtained the final output with this confidence score. (Zhang et al., 2020) proposed a flow-based model that retrieved the feature map of test samples and fed it into an additional network. Such an additional network was designed only for forecasting whether the test sample is ID or not. (Yu & Aizawa, 2019) firstly trained an encoder in a supervised way, and then two classifiers for ID samples and OOD samples were trained with an unsupervised objective, respectively. All these methods aim to redesign or introduce new layers or auxiliary networks with corresponding objectives for OOD detection. Retraining a network can be extremely time-consuming, especially when the parameter scale is substantially large. The modified objective that considers OOD detection in addition to the original task may also have side effects of degrading model performance on the original task. In addition, some methods require OOD samples as input, which imposes additional requirements on the datasets. A potential risk is that the model will achieve poor results on data beyond the distribution of the trained OOD samples. Unlike those approaches, our method does not need to make any changes (including both architecture and objective) to the original network and does not need to do any additional training, which is a plausible property for real applications.

**Network Output Transformation.** Apart from adjusting network structure or retraining the network with revised objectives, transforming network outputs to obtain the desired measure is the other classic OOD detection paradigm. The first study on network output transformation was proposed by (Hendrycks & Gimpel, 2016), which used the Maximum Softmax Probability (MSP) to measure the confidence of test samples. Intuitively, ID data is more likely to obtain high confidence from Softmax measure than OOD samples. In (Liang et al., 2017), input data were perturbed with ID-sample-friendly perturbations and the Softmax probability was re-scaled by a temperature parameter $T$; thus OOD data and ID data are more separable. (Lee et al., 2018) first generated class-conditional Gaussian distribution from middle layer outputs of the already trained network on training data and then calculated the confidence of test samples under the Mahalanobis distance measure. (Serrà et al., 2019) made assumptions about the complexity of output and input images, and they advocated estimating the complexity of the input image to impact the output for efficient OOD detection. (Liu et al., 2020) detected the OOD samples by an energy-based score function on the final output logits. Note that such energy function is different from ours as it calculates such score merely based on the output logits of the testing sample instead of comparing with stored patterns. Based on the observation that the mean activation of the OOD sample had larger variations, (Sun et al., 2021) set a threshold to clip the output of the penultimate layer, thereby reducing the output magnitude of OOD samples in the last layer. These approaches can be directly applied to OOD detection tasks without additional training, which is more practical for real-world applications. (Sehwag et al., 2021) utilize the advantage of self-supervised training. (Sun et al., 2022) calculate

all the Euclidean distance with each pattern of train sample and use the k-th sorted distance as the metric for OOD detection.

# 3 METHODOLOGY

In this section, we first introduce the preliminaries of the OOD detection task and the Hopfield Network. Then we elaborate on how to leverage the critical concepts of the Hopfield Network on OOD detection. More precisely, the energy function defined in the Modern Hopfield Network (Ramsauer et al., 2020) is introduced as the basis of our store-then-compare OOD detection paradigm. A simplified energy function is further proposed to reduce the memory demand, which is of high computational efficiency and free of hyperparameters. Finally, we compare the difference in pattern choice, i.e., patterns derived from the penultimate layer outputs versus final logits. We also conduct in-depth theoretical analyses of our method.

## 3.1 OUT-OF-DISTRIBUTION DETECTION

A neural network $f$ aims to learn a mapping function from a training sample $\boldsymbol{x}$ to its corresponding label $\boldsymbol{y}$ as $\boldsymbol{y} = f(\boldsymbol{x}; \boldsymbol{\theta})$ with parameter $\boldsymbol{\theta}$. Then a testing sample $\boldsymbol{x}'$ is fed into the trained network $f$ for the prediction $\boldsymbol{y}'$. When $\boldsymbol{x}'$ and training sample $\boldsymbol{x}$ are from the same data distribution, then $\boldsymbol{x}'$ is called an ID sample; otherwise, it is regarded as an OOD sample. Prediction results for OOD samples in turn fail to be meaningful. More severely, blindly classifying OOD samples into any existing class may raise fatal risks in safety-critical scenarios. Thus, the OOD detection task is to design a measure function $D(f; \boldsymbol{x}')$ that allows OOD samples to be as clearly distinguishable from ID samples as possible. Eventually, OOD detection can be formulated as follows:

$$\boldsymbol{x}' \sim \begin{cases} \text{OOD} & \text{if } D(f; \boldsymbol{x}') = 0 \\ \text{ID} & \text{if } D(f; \boldsymbol{x}') = 1. \end{cases} \tag{1}$$

## 3.2 HOPFIELD NETWORK

Hopfield Network (Hopfield, 1984; Krotov & Hopfield, 2016; Ramsauer et al., 2020) can store and retrieve continuous patterns. By minimizing the predefined energy function, it can gradually update the input test pattern $\boldsymbol{\xi} \in \mathbb{R}^{d \times 1}$ to a certain converged pattern that is similar to one of the stored patterns. We denote all stored patterns as a stored pattern set $\boldsymbol{S} \in \mathbb{R}^{d \times N}$ with each column $\boldsymbol{s}_j \in \mathbb{R}^{d \times 1}$ representing one specific stored pattern, and $N$ is the total number of the stored pattern. Here $d$ is the dimension of patterns. Thus, the energy function aims to guide the updating procedure in the Modern Hopfield Network can be written as:

$$\text{Energy} = -\text{LSE}\left(\beta, \boldsymbol{\xi}^T \boldsymbol{S}\right) + \frac{1}{2}\boldsymbol{\xi}^T \boldsymbol{\xi} + c \tag{2}$$

$$\text{LSE}(\beta, \boldsymbol{e}) = \beta^{-1} \log\left(\sum_{j=1}^{N} \exp\left(\beta e_j\right)\right), \tag{3}$$

where LSE denotes the *log-sum-exp* function and is defined in Eq. 3. $\beta$ and $c$ are two constant. The vector $\boldsymbol{e}$ denotes $\boldsymbol{\xi}^T \boldsymbol{S}$, where $e_j$ represents the inner product of the input test pattern $\boldsymbol{\xi}$ and the $j$-th stored pattern $\boldsymbol{s}_j$. The second term $\boldsymbol{\xi}^T \boldsymbol{\xi}$ on the right of Eq. 2 serves as a regularization on the magnitude of $\boldsymbol{\xi}$. Revisiting the energy function, we can find that it is essentially a measure that depicts the similarity between training patterns and testing patterns.

## 3.3 **HE**: OOD DETECTION WITH MODERN HOPFIELD ENERGY

As described above, it is obvious that the energy function of Modern Hopfield Network can be an appropriate candidate for measuring the discrepancy between OOD instances and ID instances. We denote all stored patterns for class $i$ as a stored pattern set $\boldsymbol{S}_i \in \mathbb{R}^{d \times N_i}$ with each column $\boldsymbol{s}_{ij} \in \mathbb{R}^{d \times 1}$ representing one specific stored pattern, and $N_i$ is the total number of the stored patterns within class $i$. More precisely, we preserve the penultimate layer outputs of ID training samples as stored patterns: for each class $i$, a corresponding stored pattern set $\boldsymbol{S}_i$ is derived from the penultimate layer

outputs of ID training samples that are correctly classified by the network within this class. When testing a new sample, we can obtain its penultimate layer output $\boldsymbol{\xi}$ as well as its prediction result, e.g., class $i$, from the trained model. Then, we just need to conduct the calculation between $\boldsymbol{\xi}$ and the corresponding stored pattern set $\boldsymbol{S_i}$ to obtain the similarity score.

Notice that, there is the magnitude regularization item $\boldsymbol{\xi}^T\boldsymbol{\xi}$ in the energy function Eq. 2. Since it is introduced to prevent the input pattern from scaling-up during the pattern updating process of Modern Hopfield Network, it is not necessary when measuring the discrepancy between the input pattern and the stored patterns. Thus we omit such a term as the magnitude of the input pattern $\boldsymbol{\xi}$ can also provide some information for the ID and OOD discrimination. Also, we omit the constant $c$ in Eq. 2 because the constant in the measure function does not change the OOD detection result. In summary, the Hopfield energy-based OOD detection measure can be denoted as ($i$ denotes the classfication result of $\xi$ by the already trained model):

$$\textbf{HE}(\boldsymbol{\xi}) = \text{LSE}\left(\beta, \boldsymbol{\xi}^T\boldsymbol{S_i}\right). \tag{4}$$

Note that, Eq. 4 measures the similarity instead of discrepancy, which means the higher score indicates ID data while lower score as OOD data. It is worth mentioning that, different from most traditional OOD detection methods that only consider the information from the input test sample itself with the trained network, we leverage the information from all training samples of the predicted class to make the comparison for better OOD detection.

### 3.4 **SHE**: OOD DETECTION WITH SIMPLIFIED HOPFIELD ENERGY

Although HE has the theoretical foundation from the Modern Hopfield network and is proven to be effective through empirical evaluation, the need to store patterns of all correctly classified training samples may prevent it from generalization to real-world applications. Particularly when the scale of the dataset is extremely large or the latent representation is ultra-high dimensional, it will impose a considerable burden on the storage and the computation. During the evaluation, we discover that the hyperparameter $\beta$ in Eq. 4 should be small enough in case that there is any element $e_{ij}$ (denotes the inner product between the testing pattern and the $j$-th stored pattern of class $i$) with extra large value, which will degrade the robustness of OOD detection with large $\beta$. When $\beta$ is relatively small, we can transform Eq. 4 with Taylor series (here we use two Taylor series as $\exp(x) \approx 1 + x$ and $\log(1 + x) \approx x$) by:

$$\begin{aligned}
\text{LSE}(\beta, \boldsymbol{e}_i) &= \frac{1}{\beta} \log\left(\sum_{j=1}^{N_i} \exp\left(\beta e_{ij}\right)\right) \approx \frac{1}{\beta} \log\left(\sum_{j=1}^{N_i} \left(1 + \beta e_{ij}\right)\right) \\
&= \frac{1}{\beta} \log\left(N_i + \sum_{j=1}^{N_i} \beta e_{ij}\right) = \frac{1}{\beta} \log N_i \left(1 + \beta \frac{\sum_{j=1}^{N_i} e_{ij}}{N_i}\right) \\
&= \frac{1}{\beta} \log N_i + \frac{1}{\beta} \log\left(1 + \beta \boldsymbol{\xi}^T \frac{\sum_{j=1}^{N_i} s_{ij}}{N_i}\right).
\end{aligned} \tag{5}$$

All test samples that are predicted to be class $i$ share the same stored patterns size $N_i$, so the first term $\beta^{-1} \log N_i$ remains the same for them and can be regarded as a constant. Thus, we ignore the first term, and the measure function Eq. 4 can be simplified from the LSE function (Eq. 3) to the inner product of $\boldsymbol{\xi}$ and $\bar{\boldsymbol{S_i}}$ because $\text{LSE}(\beta, \boldsymbol{e}_i)$ is positively related to $\boldsymbol{\xi}^T\bar{\boldsymbol{S_i}}$:

$$\textbf{SHE}(\xi) = \boldsymbol{\xi}^T \bar{\boldsymbol{S_i}}. \tag{6}$$

Here $\bar{\boldsymbol{S_i}}$ is defined as:

$$\bar{\boldsymbol{S_i}} = \frac{1}{N_i} \sum_{j=1}^{N_i} \boldsymbol{s}_{ij}. \tag{7}$$

We can interpret SHE from another perspective: $\bar{\boldsymbol{S_i}}$ can be viewed as the average of vectors from the stored pattern set $\boldsymbol{S}_i$. In other words, a stored pattern set $\boldsymbol{S}_i$ is degraded to a representation vector $\bar{\boldsymbol{S_i}}$. Considering the redundancy of patterns that frequently appears in deep learning and samples from

the same class usually have similar patterns, it is reasonable to prune the stored pattern set into a single average vector, which is also shown in Figure. 1 (right). It enables us to eliminate the memory overhead of storing a large amount of ID data patterns, and further reduces the computational cost as well. Besides, the only hyperparameter $\beta$ also disappears, indicating that we do not need to tune any hyperparameter. In summary, SHE is highly efficient regarding both storage and computation and does not have any hyper parameter to tune. Such properties can indeed facilitate the deployment of SHE in practice.

### 3.5 Penultimate Layer versus Logits Layer for Pattern Storage

In this section, we analyze the benefits of choosing outputs from the penultimate layer, compared with the logits layer, as the pattern.

**Intuitive Explanation.** Given that most OOD detection methods usually use the output logits, one interesting question is how to choose the layer output as the patterns, the penultimate layer output, or the output logits. Note that, when we calculate the energy function by Eq. 6, the stored pattern $\bar{S}_i$ comes from the same category $i$ as the testing pattern $\xi$ is classified. Therefore, when we use the output logits as patterns, no matter $\xi$ comes from an ID or OOD sample, its maximum value position of logits is always the same as the maximum value position of logits of $\bar{S}_i$ by design, which is the category index of $\xi$. We call such alignment of maximum value position of logits as "Peak Alignment" which leads to a high energy function score for the OOD pattern more easily. It in turn raises the difficulty of discriminating ID and OOD samples. However, when we use the penultimate layer output as patterns, there is no such "Peak Alignment" effect between $\xi$ and $\bar{S}_i$ since the value of the penultimate layer output is not so concentrated distributed, promoting the similarity score calculated from energy function more separable for ID and OOD patterns as shown in Figure. 2. Moreover, we provide the theoretical analysis in the Appendix B.

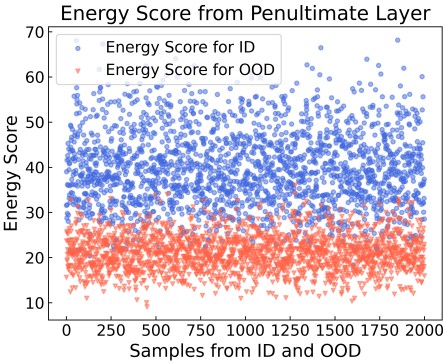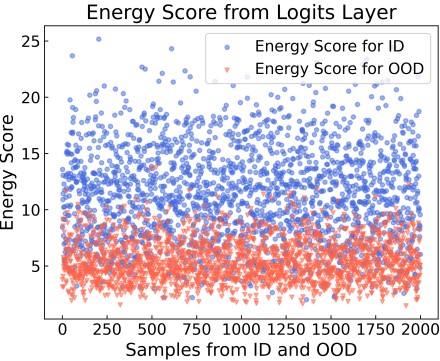

Figure 2: Distribution of the Hopfield Energy Score calculated from 2,000 ID and 2,000 OOD samples, the pattern is derived from penultimate layer (left) and output logits (right), respectively. When using penultimate layer, the score can be distinguished more for ID and OOD samples. The ID and OOD is CIFAR10 (Krizhevsky et al., 2009) and SVHN (Netzer et al., 2011) respectively, and the backbone network is ResNet18.

## 4 Experiments

In this section, we conduct experiments on nine OOD detection datasets with three backbone networks and two ID datasets to evaluate the performance of our methods.

### 4.1 Dataset

There are two types of datasets in the experiments: The in-distribution (ID) dataset and the Out-of-distribution (OOD) dataset. The former is only utilized during the training procedure, while the latter serves to test models and does not contain any ID dataset sample.

**ID Dataset.** CIFAR10, CIFAR100 and ImageNet-1k are three ID datasets in our experiments. CIFAR10 (Krizhevsky et al., 2009) is composed of 60,000 images with 10 categories, each containing 5,000 training images and 1,000 testing images. CIFAR100 Krizhevsky et al. (2009) consists of 60,000 images with 100 categories with 500 training images and 100 testing images for each class. ImageNet-1k (Deng et al., 2009) is composed of 1,350,000 images 1000 different object categories, each containing 1,300 training images and 50 testing images.

**OOD Dataset.** There are nine OOD dataset for evaluation, including SVHN (Netzer et al., 2011), LSUN-C (Yu et al., 2015) (crop) and LSUN-R (Yu et al., 2015) (resize), iSUN (Xu et al., 2015), Places (Zhou et al., 2017), DTD (Cimpoi et al., 2014), SUN (Xiao et al., 2010), iNaturalist (Van Horn et al., 2018), Tiny-Imagenet (resize) (Deng et al., 2009). Details can be found in the original papers.

## 4.2 EXPERIMENT SETTINGS

**Backbone Network.** We choose ResNet18 (He et al., 2016), ResNet34 (He et al., 2016) and WRN40-2 (Zagoruyko & Komodakis, 2016) as our backbone networks, which are trained on the ID dataset CIFAR10 and CIFAR100, respectively. For ImageNet, we choose ResNet50 (He et al., 2016) as our backbone network.

**Baseline Methods.** To evaluate the performance of our proposed design, We also conduct experiments on the Softmax-based approach "MSP" (Hendrycks & Gimpel, 2016) and other excellent methods, "Energy" (Liu et al., 2020) , "ODIN" (Liang et al., 2017), "Mahalanobis" (Lee et al., 2018) and "ReAct" (Sun et al., 2021), the "ReAct" here is combined with Energy as described in (Sun et al., 2021) and is the state-of-the-art method before.

**Evaluation Metrics.** The evaluation metrics are: (1) the False Positive Rate (FPR95) of OOD samples when the True Positive Rate of in-distribution samples is at 95%; (2) the area under the receiver operating characteristic curve (AUROC). Models with smaller FPR95 and higher AUROC results are regarded as effective. All experimental values are expressed as percentages (%) and the bolded numbers (sometimes colored with gray cell background) denote the best result.

## 4.3 EXPERIMENTAL RESULT

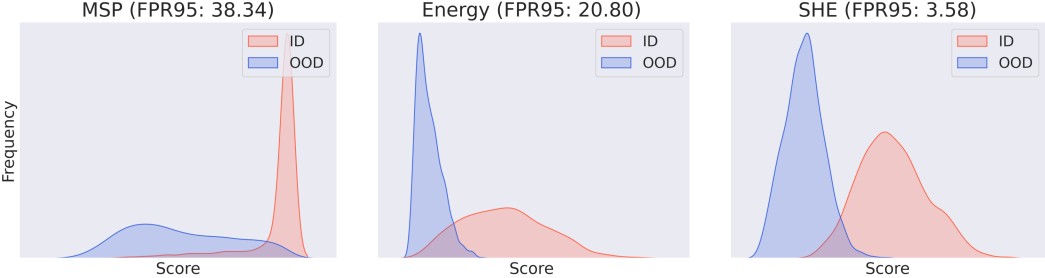

Figure 3: Confidence distribution of ID data and OOD data calculated from ResNet18. ID/OOD refers to CIFAR10 and SUN (Xiao et al., 2010), performance is compared with MSP (Hendrycks & Gimpel, 2016) and Energy (Liu et al., 2020).

### 4.3.1 OVERALL RESULTS OF **SHE**.

In this section, we demonstrate the effectiveness of our method "SHE" through extensive experiments. The experimental results on nine OOD datasets are organized in Table 1 taking CIFAR10 as ID training data. As shown in Table 1, MSP has the worse results, while Energy has a lot of improvement, illustrating the potential of energy-based solutions for OOD detection. Our approach SHE obtains almost all (26/27) of the best performance on nine OOD datasets for three different backbones. More precisely, our approach reduces the average FPR95 by 16.81% for ResNet18, 6.29% for ResNet34, and 12.18% for WRN40-2 compared with the best baseline. Our approach performs well when CIFAR100 or ImageNet-1k is choosed as the ID training data while with some limitation, the detailed table 5 (CIFAR100) and table 7 (ImageNet-1k) is put in the Appendix A. We also illustrate Figure 3 to demonstrate model performance directly. Under the peak of ID score

Table 1: OOD detection performance of **SHE** using CIFAR10 as ID dataset.

| Backbone Network | OOD Datasets | Methods | | | | | | | |
|---|---|---|---|---|---|---|---|---|---|
| | | MSP | | Energy | | ReAct | | **SHE (Ours)** | |
| | | FPR95 ($\downarrow$) | AUROC ($\uparrow$) | FPR95 ($\downarrow$) | AUROC ($\uparrow$) | FPR95 ($\downarrow$) | AUROC ($\uparrow$) | FPR95 ($\downarrow$) | AUROC ($\uparrow$) |
| ResNet18 | SVHN | 74.99 | 85.83 | 51.81 | 91.05 | 54.35 | 90.52 | **5.87** | 98.74 |
| | LSUN-C | 44.59 | 94.36 | 14.91 | 97.21 | 14.73 | 97.20 | **7.94** | 98.45 |
| | LSUN-R | 38.93 | 94.75 | 14.98 | 97.45 | 14.51 | 97.49 | **6.67** | 98.42 |
| | iSUN | 35.82 | 95.24 | 11.99 | 97.76 | 11.78 | 97.80 | **4.16** | 98.85 |
| | Places | 39.16 | 94.39 | 21.06 | 96.57 | 17.36 | 97.14 | **6.31** | 98.70 |
| | DTD | 54.93 | 89.67 | 54.58 | 86.95 | 48.99 | 91.42 | **32.02** | 89.11 |
| | Tiny Imagenet | 44.49 | 93.76 | 27.76 | 95.83 | 28.06 | 95.81 | **11.81** | 97.86 |
| | SUN | 38.34 | 94.65 | 20.80 | 96.80 | 15.08 | 97.44 | **3.58** | 99.24 |
| | iNaturalist | 68.40 | 88.80 | 65.16 | 89.59 | 52.10 | 92.80 | **27.32** | 95.02 |
| | Average | 48.85 | 92.38 | 31.45 | 94.36 | 28.55 | 95.29 | **11.74** | 97.15 |
| ResNet34 | SVHN | 38.67 | 95.27 | 14.87 | 97.48 | 15.36 | 97.38 | **3.16** | 99.34 |
| | LSUN-C | 27.27 | 96.43 | 6.05 | 98.62 | 6.77 | 98.53 | **2.37** | 99.44 |
| | LSUN-R | 34.53 | 95.38 | 9.25 | 98.15 | 8.08 | 98.29 | **5.73** | 98.71 |
| | iSUN | 33.15 | 95.56 | 8.69 | 98.21 | 8.06 | 98.32 | **4.13** | 99.04 |
| | Places | 32.99 | 95.34 | 12.37 | 97.63 | 11.81 | 97.64 | **2.86** | 99.32 |
| | DTD | 37.38 | 94.27 | 26.10 | 94.63 | 21.07 | 95.98 | **12.76** | 96.57 |
| | Tiny Imagenet | 44.26 | 93.31 | 24.17 | 95.80 | 21.91 | 95.98 | **16.66** | 97.03 |
| | SUN | 29.86 | 95.77 | 10.03 | 98.02 | 9.85 | 97.99 | **1.47** | 99.63 |
| | iNaturalist | 22.64 | 96.73 | 11.07 | 97.55 | 7.85 | 98.17 | **5.05** | 99.08 |
| | Average | 33.42 | 95.34 | 13.62 | 97.34 | 12.31 | 97.59 | **6.02** | 98.68 |
| WRN40-2 | SVHN | 54.05 | 92.34 | 30.80 | 95.09 | 36.93 | 93.96 | **12.58** | 97.70 |
| | LSUN-C | 49.58 | 92.85 | **23.91** | 96.03 | 25.89 | 95.46 | 32.33 | 93.94 |
| | LSUN-R | 42.86 | 94.04 | 15.91 | 97.34 | 14.91 | 97.42 | **9.19** | 98.10 |
| | iSUN | 44.43 | 94.01 | 16.76 | 97.23 | 16.77 | 97.26 | **8.32** | 98.30 |
| | Places | 42.88 | 94.01 | 16.98 | 97.06 | 20.30 | 96.44 | **5.27** | 98.84 |
| | DTD | 40.00 | 94.25 | 27.97 | 95.00 | 25.12 | 95.86 | **13.26** | 96.10 |
| | Tiny Imagenet | 55.44 | 91.37 | 38.77 | 93.87 | 38.22 | 94.34 | **24.20** | 95.56 |
| | SUN | 41.04 | 94.49 | 15.31 | 97.41 | 18.80 | 96.76 | **3.51** | 99.18 |
| | iNaturalist | 41.65 | 94.64 | 36.85 | 95.30 | 33.19 | 95.52 | **5.05** | 98.84 |
| | Average | 45.77 | 93.56 | 24.81 | 96.04 | 25.57 | 95.89 | **12.63** | 97.40 |

Table 2: OOD detection performance comparison of HE and SHE. All values are averaged over the nine OOD datasets described in Section 4.1. The detailed Table 9 is displayed in the Appendix A.

| Backbone Network | CIFAR10 | | | | CIFAR100 | | | |
|---|---|---|---|---|---|---|---|---|
| | HE | | SHE | | HE | | SHE | |
| | FPR95 ($\downarrow$) | AUROC ($\uparrow$) | FPR95 ($\downarrow$) | AUROC ($\uparrow$) | FPR95 ($\downarrow$) | AUROC ($\uparrow$) | FPR95 ($\downarrow$) | AUROC ($\uparrow$) |
| ResNet18 | 11.77 | 97.13 | **11.74** | 97.15 | **48.73** | 87.16 | 48.86 | 87.14 |
| ResNet34 | 6.03 | 98.69 | **6.02** | 98.68 | **63.23** | 81.42 | 63.28 | 81.48 |
| WRN40-2 | 12.69 | 97.35 | **12.63** | 97.40 | 37.22 | 91.66 | **36.16** | 92.17 |

distribution (colored red), the less data in blue indicates the better capability in distinguish OOD data and ID data. It can be seen that SHE can obtain the minimum blue areas under the red peak among three methods.

### 4.3.2 **SHE** VERSUS **HE**

Notice that SHE is derived from HE that is inspired by Modern Hopfield Network, we make a comparison between them. We can discover from Table 2 that SHE and HE are competitive, indicating that the patterns of samples from the same class are similar and contain redundancy for OOD detection. As mentioned in chapter 3, **HE** detects OOD samples via the energy function as shown in Eq. 4. We evaluate the feasibility of **HE** and show the results in Table 2, which demonstrates that the energy function (Eq. 4) is able for OOD detection. For SHE, memory cost is reduced to the number of classes instead of the number of samples and the energy-based measure can be simplified into frequently used inner products without any hyperparameter, which is elegant and efficient. We also provide detailed comparison results on nine OOD detection in Table 9, proving that in most cases SHE can achieve similar performance to HE.

### 4.3.3 PERTURBATIONS ON **SHE**

Adding perturbations to input samples for OOD detection is proposed by (Liang et al., 2017), which can be formulated as follows:

$$\tilde{\boldsymbol{x}}' = \boldsymbol{x}' + \varepsilon\,\mathrm{sign}\left(\nabla_{\boldsymbol{x}'}\log S(\boldsymbol{x}')\right), \tag{8}$$

where $\boldsymbol{x}'$ denotes the testing sample to be detected while $\tilde{\boldsymbol{x}}'$ is the perturbed one. $S(\boldsymbol{x}')$ is the maximum Softmax probability of network outputs, and $\varepsilon$ is the perturbation magnitude. Such

perturbation-based methods have been proved to be more beneficial for ID data than OOD data, and are adopted in extensive studies (Lee et al., 2018; Hsu et al., 2020; DeVries & Taylor, 2018). To verify whether perturbation applies to SHE, we also carry out the experiments and organize the results in Table 3. It demonstrates that introducing perturbations can be beneficial to SHE as the performance of SHE could be further improved with perturbations.

Note that, the perturbation requires an additional "forward-backward" procedure to retrieve the gradient, leading to an increase in computational complexity and time. As a comparison, we record the time overhead before and after adding perturbation to SHE: when we choose ResNet34 as the backbone with CIFAR100 as ID data and TinyImagenet as OOD data, the consuming time raises from $35.61s$ to $105.88s$ that is around 3 times. Therefore, a balance between efficiency and computational overhead needs to be considered when adopting perturbation. Nevertheless, SHE is always efficient and effective, which can be combined with perturbations to achieve even higher accuracy.

Table 3: OOD detection performance comparison of SHE and SHEP (i.e., SHE + Perturbation). All values are averaged over the nine OOD datasets described in Section 4.1. The detailed Table 10 is displayed in the Appendix.

| Backbone Network | CIFAR10 | | | | CIFAR100 | | | |
|---|---|---|---|---|---|---|---|---|
| | SHE | | SHEP | | SHE | | SHEP | |
| | FPR95 ($\downarrow$) | AUROC ($\uparrow$) | FPR95 ($\downarrow$) | AUROC ($\uparrow$) | FPR95 ($\downarrow$) | AUROC ($\uparrow$) | FPR95 ($\downarrow$) | AUROC ($\uparrow$) |
| ResNet18 | 11.74 | 97.15 | **6.52** | 98.46 | 48.86 | 87.14 | **43.62** | 88.40 |
| ResNet34 | 6.02 | 98.68 | **4.26** | 99.09 | 63.28 | 81.48 | **57.54** | 84.50 |
| WRN40-2 | **12.63** | 97.40 | 13.69 | 96.97 | **36.16** | 92.17 | 41.42 | 90.26 |

Table 4: OOD detection performance comparison deriving pattern from penultimate and logits layer. All values are averaged over the nine OOD datasets described in Section 4.1. The detailed Table 11 is displayed in the Appendix.

| Backbone Network | CIFAR10 | | | | CIFAR100 | | | |
|---|---|---|---|---|---|---|---|---|
| | Logits | | Penultimate | | Logits | | Penultimate | |
| | FPR95 ($\downarrow$) | AUROC ($\uparrow$) | FPR95 ($\downarrow$) | AUROC ($\uparrow$) | FPR95 ($\downarrow$) | AUROC ($\uparrow$) | FPR95 ($\downarrow$) | AUROC ($\uparrow$) |
| ResNet18 | 31.70 | 94.32 | **11.74** | 97.15 | 76.94 | 77.77 | **48.86** | 87.14 |
| ResNet34 | 13.92 | 97.33 | **6.02** | 98.68 | 80.75 | 77.88 | **63.28** | 81.48 |
| WRN40-2 | 24.28 | 96.02 | **12.63** | 97.40 | 55.99 | 88.12 | **36.16** | 92.17 |

### 4.3.4 PENULTIMATE LAYER VS. LOGITS LAYER

Note that all the stored patterns or test patterns in our approach are obtained from the penultimate layer output of neural networks. But in most methods, logits layer (i.e., the last layer) output is adopted for the confidence computation (Hendrycks & Gimpel, 2016; Liang et al., 2017; Liu et al., 2020; Sun et al., 2021). To verify the significance of layer selection, we also provide an experimental evaluation on SHE that chooses patterns from the penultimate layer outputs or final logits, respectively, for comparison. The average results on nine datasets are presented in Table 4. We can conclude that our approach gets much better performance when using the penultimate layer output, instead of final logits, as patterns to apply SHE.

## 5 CONCLUSION

In this paper, we propose a novel approach named **HE** for OOD detection based on a new "store-then-compare" paradigm. The key idea is to store patterns of ID data and then leverage the energy function defined in the Modern Hopfield Network (Ramsauer et al., 2020) for measuring the similarity between the new test patterns and the stored ID patterns. To reduce storage and computational overhead, we simplify the energy function with the theoretical analysis by appropriate approximations and obtain the simplified approach named **SHE**. In addition to the great efficiency and effectiveness, **SHE** does not have any hyperparameters to tune, which is more convenient than most OOD detection methods with cumbersome hyperparameters tuning. Also, different from most OOD detection methods focusing on final output logits, we find that the penultimate layer output, rather than the final output logits, is more suitable to be used as patterns in our approach for OOD detection. The conducted evaluations demonstrate the superiority of our proposed simple yet effective approach on nine widely-used OOD datasets.

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

## A  EXPERIMENTAL RESULTS.

Table 5: OOD detection performance of **SHE** using CIFAR100 as ID dataset.

| Backbone Network | OOD Datasets | Methods | | | | | | | |
|---|---|---|---|---|---|---|---|---|---|
| | | MSP | | Energy | | ReAct | | **SHE (Ours)** | |
| | | FPR95 (↓) | AUROC (↑) | FPR95 (↓) | AUROC (↑) | FPR95 (↓) | AUROC (↑) | FPR95 (↓) | AUROC (↑) |
| ResNet18 | SVHN | 61.61 | 87.54 | 46.01 | 92.74 | 53.07 | 89.90 | 12.56 | 97.64 |
| | LSUN-C | 63.09 | 84.66 | 43.98 | 91.68 | **19.95** | 96.67 | 27.93 | 95.15 |
| | LSUN-R | 89.15 | 62.05 | 77.84 | 78.65 | **60.06** | 88.66 | 67.36 | 86.59 |
| | iSUN | 88.25 | 66.06 | 76.66 | 80.16 | 58.08 | 89.78 | **53.33** | 90.10 |
| | Places | 87.14 | 74.96 | 85.84 | 74.81 | 75.66 | 84.94 | **51.14** | 88.03 |
| | DTD | 91.47 | 68.50 | 93.16 | 60.86 | 86.07 | 78.30 | **65.30** | 68.16 |
| | Tiny Imagenet | 84.02 | 69.32 | 72.28 | 81.33 | 58.44 | 89.66 | **58.25** | 89.13 |
| | SUN | 89.42 | 72.85 | 88.63 | 73.08 | 79.69 | 83.56 | **44.48** | 90.62 |
| | iNaturalist | 92.95 | 67.60 | 94.31 | 68.45 | 85.15 | 81.73 | **59.42** | 78.85 |
| | Average | 83.01 | 72.62 | 75.41 | 77.97 | 64.02 | 87.02 | **48.86** | 87.14 |
| ResNet34 | SVHN | 58.43 | 88.65 | 42.75 | 93.08 | 26.46 | 94.43 | **14.38** | 97.32 |
| | LSUN-C | 78.25 | 81.79 | 69.77 | 87.41 | 35.10 | 94.98 | **33.04** | 94.29 |
| | LSUN-R | 92.01 | 62.75 | 85.38 | 74.81 | **72.59** | 83.30 | 80.26 | 76.60 |
| | iSUN | 90.54 | 65.62 | 84.92 | 75.92 | **71.05** | 84.45 | 72.51 | 80.20 |
| | Places | 88.13 | 75.94 | 88.77 | 73.61 | **59.71** | 89.05 | 77.51 | 77.63 |
| | DTD | 87.83 | 73.33 | 88.56 | 69.91 | **60.17** | 88.35 | 68.59 | 72.49 |
| | Tiny Imagenet | 88.94 | 67.08 | 84.08 | 75.83 | **71.73** | 84.11 | 79.07 | 76.59 |
| | SUN | 91.84 | 72.72 | 92.62 | 69.75 | **65.09** | 88.45 | 77.94 | 77.88 |
| | iNaturalist | 88.41 | 79.13 | 90.03 | 79.67 | **54.06** | 91.37 | 66.22 | 80.29 |
| | Average | 84.93 | 74.11 | 80.76 | 77.78 | **57.33** | 88.72 | 63.28 | 81.48 |
| | SVHN | 69.45 | 84.07 | 52.60 | 92.37 | 52.05 | 91.86 | **19.15** | 96.54 |
| | LSUN-C | 59.67 | 85.31 | 33.43 | 94.35 | 31.13 | 94.10 | **25.94** | 95.47 |
| | LSUN-R | 87.28 | 64.19 | 72.26 | 82.67 | **71.56** | 84.00 | 75.04 | 82.34 |
| | iSUN | 86.06 | 67.06 | 69.92 | 84.04 | 66.83 | 85.68 | **63.44** | 86.05 |
| | Places | 68.05 | 82.24 | 47.22 | 90.26 | 45.28 | 90.33 | **24.26** | 95.13 |
| | DTD | 65.29 | 80.26 | 47.35 | 87.92 | 36.56 | 89.82 | **25.17** | 93.27 |
| | Tiny Imagenet | 80.42 | 70.35 | 63.44 | 85.91 | **56.69** | 88.64 | 62.71 | 86.66 |
| | SUN | 71.00 | 80.81 | 49.33 | 90.13 | 47.33 | 90.26 | **20.76** | 95.95 |
| | iNaturalist | 47.20 | 90.85 | 17.76 | 96.76 | 18.82 | 96.38 | **9.25** | 98.11 |
| | iNaturalist | 70.49 | 78.35 | 50.37 | 89.38 | 47.36 | 90.12 | **36.19** | 92.17 |

The **ReAct** in this table refers to 'Energy + ReAct' as described in Sun et al. (2021), as an auxillary method combined with other SOTA methods, we can also combine the **ReAct** with **SHE** (better than **Energy**) , which will also outperform **SHE** itself, the results are shown in Table. 6.

Table 6: OOD detection performance comparison of "ReAct + SHE" and "ReAct + Energy".

| FPR95 (↓) | OOD Datasets | | | | | | |
|---|---|---|---|---|---|---|---|
| Method | SVHN | LSUN-C | Place365 | DTD | SUN | iNaturalist | Avg |
| ReAct+Energy | **14.38** | **33.04** | 77.51 | 68.59 | 77.94 | 66.22 | 56.28 |
| ReAct+SHE | 23.94 | 36.79 | **62.38** | **52.53** | **56.50** | **57.28** | **48.24** |

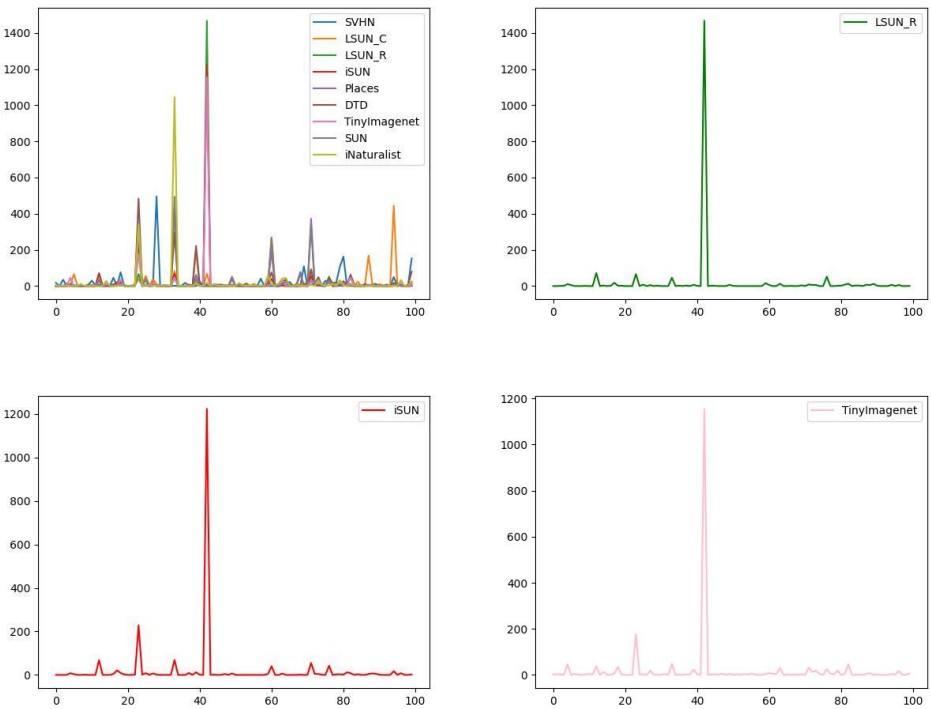

Figure 4: The distribution of output num for each class from nine OOD datasets. The backbone network is ResNet34 and the ID dataset is CIFAR100.

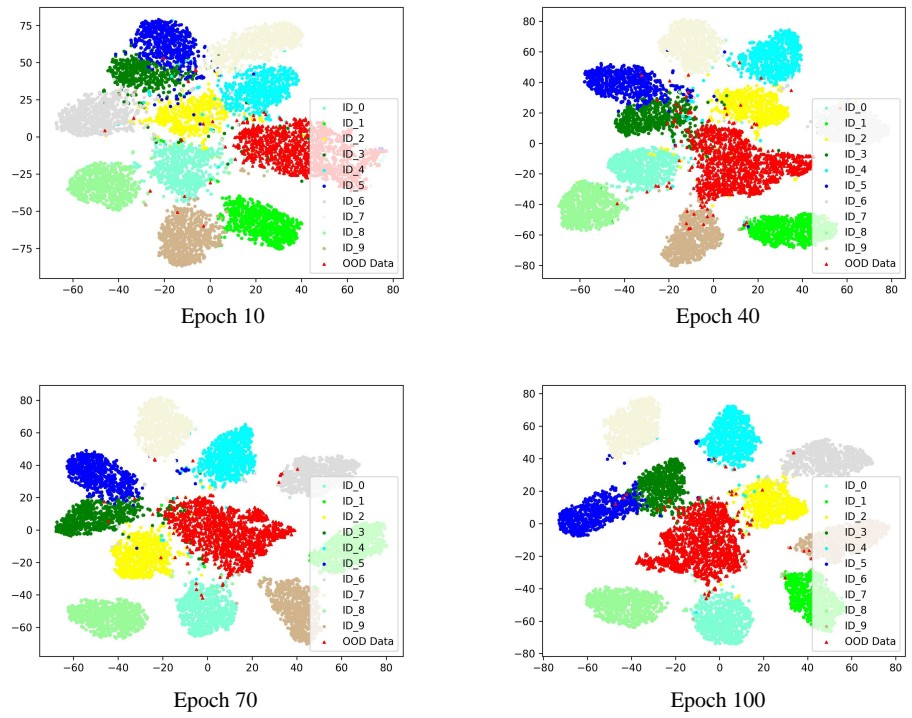

Figure 5: Visualization of the training process by t-SNE (Van der Maaten & Hinton, 2008). From the The backbone network is ResNet18, and ID data and OOD data are CIFAR10 (Krizhevsky et al., 2009) and SUN (Xiao et al., 2010) respectively.

Table 7: OOD detection performance of **SHE** using ImageNet-1k as ID dataset and ResNet50 as backbone network.

| OOD Datasets | Methods | | | | | | | | | | | |
|---|---|---|---|---|---|---|---|---|---|---|---|---|
| | MSP | | Energy | | ODIN | | Mahalanobis | | KNN | | SHE (Ours) | |
| | FPR95 (↓) | AUROC (↑) | FPR95 (↓) | AUROC (↑) | FPR95 (↓) | AUROC (↑) | FPR95 (↓) | AUROC (↑) | FPR95 (↓) | AUROC (↑) | FPR95 (↓) | AUROC (↑) |
| Places | 54.99 | 87.74 | 55.72 | 89.95 | 47.66 | 89.66 | 97.00 | 52.65 | 59.00 | 86.47 | **45.35** | 90.15 |
| DTD | 70.83 | 80.86 | 59.26 | 85.89 | 60.15 | 84.59 | 98.50 | 42.41 | 68.82 | 80.72 | **45.09** | 87.93 |
| SUN | 73.99 | 79.76 | 64.92 | 82.86 | 67.89 | 81.78 | 98.40 | 41.79 | 76.28 | 75.76 | **54.19** | 84.69 |
| iNaturalist | 68.00 | 79.61 | 53.72 | 85.99 | 50.23 | 85.62 | 55.80 | 85.01 | **11.77** | 97.07 | 34.22 | 90.18 |
| Average | 66.95 | 81.99 | 58.41 | 86.17 | 56.48 | 85.41 | 87.43 | 55.47 | 53.97 | 85.01 | **44.71** | 88.24 |

Table 8: OOD detection performance of Energy+ReAct and SHE+ReAct using ImageNet as ID dataset. From the results, ReAct+Energy is better which is limitation of our method and we will explore it in the future.

| FPR95(%) | Place365 | DTD | SUN | iNaturalist | Avg |
|---|---|---|---|---|---|
| Energy+ReAct | 20.38 | 24.20 | 33.85 | 47.30 | 31.43 |
| SHE+ReAct | 49.91 | 32.06 | 41.70 | 35.99 | 39.92 |

Table 9: OOD detection performance comparison of HE and SHE. All values are percentages. Bold numbers with gray cell are superior results. The hyperparameter $\beta$ used for **HE** is 0.01 for ResNet18, ResNet34, 0.2 for WRN40-2.

| Backbone Network | OOD Datasets | CIFAR10 | | | | CIFAR100 | | | |
|---|---|---|---|---|---|---|---|---|---|
| | | HE | | SHE | | HE | | SHE | |
| | | FPR95 (↓) | AUROC (↑) | FPR95 (↓) | AUROC (↑) | FPR95 (↓) | AUROC (↑) | FPR95 (↓) | AUROC (↑) |
| ResNet18 | SVHN | 6.31 | 98.70 | **5.87** | 98.74 | 12.97 | 97.63 | **12.56** | 97.64 |
| | LSUN-C | 7.95 | 98.45 | **7.94** | 98.45 | **27.51** | 95.24 | 27.93 | 95.15 |
| | LSUN-R | 6.93 | 98.42 | **6.67** | 98.42 | 66.81 | 86.78 | 67.36 | 86.59 |
| | iSUN | 4.13 | 98.86 | **4.16** | 98.85 | 53.24 | 90.15 | 53.33 | 90.10 |
| | Places | 6.44 | 98.66 | **6.31** | 98.70 | **50.95** | 88.17 | 51.14 | 88.03 |
| | DTD | 32.18 | 88.95 | **32.02** | 89.11 | 65.41 | 67.87 | **65.30** | 68.16 |
| | Tiny Imagenet | **11.62** | 97.86 | 11.81 | 97.86 | **57.60** | 89.31 | 58.25 | 89.13 |
| | SUN | **3.51** | 99.23 | 3.58 | 99.24 | **44.46** | 90.65 | 44.48 | 90.62 |
| | iNaturalist | **26.88** | 95.06 | 27.32 | 95.02 | 59.58 | 78.63 | **59.42** | 78.85 |
| | Average | 11.77 | 97.13 | **11.74** | 97.15 | **48.73** | 87.16 | 48.86 | 87.14 |
| ResNet34 | SVHN | 3.34 | 99.30 | **3.16** | 99.34 | **14.19** | 97.28 | 14.38 | 97.32 |
| | LSUN-C | **2.17** | 99.46 | 2.37 | 99.44 | 33.37 | 94.21 | **33.04** | 94.29 |
| | LSUN-R | **5.59** | 98.76 | 5.73 | 98.71 | **79.84** | 76.95 | 80.26 | 76.6 |
| | iSUN | 4.14 | 99.05 | **4.13** | 99.04 | 72.52 | 80.28 | **72.51** | 80.2 |
| | Places | 2.93 | 99.35 | **2.86** | 99.32 | **77.43** | 77.40 | 77.51 | 77.63 |
| | DTD | 12.84 | 96.61 | **12.76** | 96.57 | 69.75 | 71.91 | **68.59** | 72.49 |
| | Tiny Imagenet | 16.67 | 96.99 | **16.66** | 97.03 | **78.39** | 76.43 | 79.07 | 76.59 |
| | SUN | **1.42** | 99.64 | 1.47 | 99.63 | **77.53** | 78.01 | 77.94 | 77.88 |
| | iNaturalist | 5.17 | 99.07 | **5.05** | 99.08 | **66.08** | 80.31 | 66.22 | 80.29 |
| | Average | 6.03 | 98.69 | **6.02** | 98.68 | **63.23** | 81.42 | 63.28 | 81.48 |
| WRN40-2 | SVHN | 12.84 | 97.65 | **12.58** | 97.7 | 23.49 | 95.29 | **19.15** | 96.54 |
| | LSUN-C | 32.96 | 93.66 | **32.33** | 93.94 | **25.01** | 95.31 | 25.94 | 95.47 |
| | LSUN-R | **8.83** | 98.18 | 9.19 | 98.1 | 73.38 | 82.67 | 75.04 | 82.34 |
| | iSUN | **7.83** | 98.40 | 8.32 | 98.3 | 62.75 | 86.02 | 63.44 | 86.05 |
| | Places | **5.12** | 98.87 | 5.27 | 98.84 | 25.46 | 94.85 | **24.26** | 95.13 |
| | DTD | 14.44 | 95.67 | **13.26** | 96.1 | 29.94 | 91.28 | **25.17** | 93.27 |
| | Tiny Imagenet | **23.83** | 95.66 | 24.2 | 95.56 | **61.85** | 86.32 | 62.71 | 86.66 |
| | SUN | 3.58 | 99.18 | **3.51** | 99.18 | 23.36 | 95.19 | **20.76** | 95.95 |
| | iNaturalist | **4.79** | 98.86 | 5.05 | 98.84 | 9.70 | 98.03 | **9.02** | 98.11 |
| | Average | 12.69 | 97.35 | **12.63** | 97.40 | 37.22 | 91.66 | **36.16** | 92.17 |

Table 10: OOD detection performance comparison of SHE and SHEP (SHE + Perturbation). All values are percentages. Bold numbers with gray cell are superior results.

| Backbone Network | OOD Datasets | CIFAR10 | | | | CIFAR100 | | | |
|---|---|---|---|---|---|---|---|---|---|
| | | SHE | | SHEP | | SHE | | SHEP | |
| | | FPR95 (↓) | AUROC (↑) | FPR95 (↓) | AUROC (↑) | FPR95 (↓) | AUROC (↑) | FPR95 (↓) | AUROC (↑) |
| ResNet18 | SVHN | **5.87** | 98.74 | 11.11 | 97.92 | **12.56** | 97.64 | 34.61 | 93.62 |
| | LSUN-C | **7.94** | 98.45 | 8.23 | 98.32 | 27.93 | 95.15 | **26.97** | 95.31 |
| | LSUN-R | 6.67 | 98.42 | **3.32** | 99.15 | 67.36 | 86.59 | **66.96** | 83.84 |
| | iSUN | 4.16 | 98.85 | **2.69** | 99.30 | **53.33** | 90.10 | 53.54 | 87.61 |
| | Places | 6.31 | 98.70 | **1.89** | 99.49 | 51.14 | 88.03 | **36.83** | 90.71 |
| | DTD | 32.02 | 89.11 | **17.04** | 95.14 | 65.30 | 68.16 | **52.17** | 76.23 |
| | Tiny Imagenet | 11.81 | 97.86 | **5.70** | 98.76 | 58.25 | 89.13 | **55.36** | 88.07 |
| | SUN | 3.58 | 99.24 | **0.91** | 99.72 | 44.48 | 90.62 | **30.17** | 92.93 |
| | iNaturalist | 27.32 | 95.02 | **7.76** | 98.32 | 59.42 | 78.85 | **35.96** | 87.32 |
| | Average | 11.74 | 97.15 | **6.52** | 98.46 | 48.86 | 87.14 | **43.62** | 88.40 |
| ResNet34 | SVHN | **3.16** | 99.34 | 8.04 | 98.49 | **14.38** | 97.32 | 42.71 | 91.2 |
| | LSUN-C | **2.37** | 99.44 | 3.40 | 99.31 | **33.04** | 94.29 | 33.57 | 93.62 |
| | LSUN-R | 5.73 | 98.71 | **2.79** | 99.31 | 80.26 | 76.6 | **78.51** | 77.35 |
| | iSUN | 4.13 | 99.04 | **2.03** | 99.42 | 72.51 | 80.2 | **70.73** | 80.6 |
| | Places | 2.86 | 99.32 | **1.52** | 99.62 | 77.51 | 77.63 | **62.41** | 83.34 |
| | DTD | 12.76 | 96.57 | **7.05** | 98.21 | 68.59 | 72.49 | **54.34** | 81.4 |
| | Tiny Imagenet | 16.66 | 97.03 | **9.76** | 98.17 | 79.07 | 76.59 | **74.25** | 79.64 |
| | SUN | 1.47 | 99.63 | **0.87** | 99.77 | 77.94 | 77.88 | **60.77** | 84.37 |
| | iNaturalist | 5.05 | 99.08 | **2.91** | 99.49 | 66.22 | 80.29 | **40.56** | 89.02 |
| | Average | 6.02 | 98.68 | **4.26** | 99.09 | 63.28 | 81.48 | **57.54** | 84.50 |
| WRN40-2 | SVHN | **12.58** | 97.7 | 20.37 | 95.99 | **19.15** | 96.54 | 37.69 | 92.09 |
| | LSUN-C | **32.33** | 93.94 | 43.62 | 89.08 | **25.94** | 95.47 | 34.04 | 93.57 |
| | LSUN-R | 9.19 | 98.1 | **8.84** | 98.27 | **75.04** | 82.34 | 79.11 | 78.58 |
| | iSUN | 8.32 | 98.3 | **8.01** | 98.43 | **63.44** | 86.05 | 67.84 | 82.55 |
| | Places | 5.27 | 98.84 | **4.96** | 98.94 | **24.26** | 95.13 | 28.54 | 94.47 |
| | DTD | 13.26 | 96.1 | **10.57** | 97.23 | **25.17** | 93.27 | 25.31 | 93.94 |
| | Tiny Imagenet | 24.2 | 95.56 | **19.91** | 96.34 | **62.71** | 86.66 | 66.18 | 83.84 |
| | SUN | **3.51** | 99.18 | 3.71 | 99.19 | **20.76** | 95.95 | 24.39 | 95.27 |
| | iNaturalist | 5.05 | 98.84 | **3.24** | 99.27 | **9.02** | 98.11 | 9.68 | 98.05 |
| | Average | **12.63** | 97.40 | 13.69 | 96.97 | **36.16** | 92.17 | 41.42 | 90.26 |

Table 11: OOD detection performance comparison deriving pattern from penultimate and logits layer. All values are percentages. Bold numbers with gray cell are superior results.

| Backbone Network | Backbone Network | CIFAR10 | | | | CIFAR100 | | | |
|---|---|---|---|---|---|---|---|---|---|
| | | Logits | | Penultimate | | Logits | | Penultimate | |
| | | FPR95 (↓) | AUROC (↑) | FPR95 (↓) | AUROC (↑) | FPR95 (↓) | AUROC (↑) | FPR95 (↓) | AUROC (↑) |
| ResNet18 | SVHN | 54.81 | 90.61 | **5.87** | 98.74 | 49.30 | 92.31 | **12.56** | 97.64 |
| | LSUN-C | 15.09 | 97.15 | **7.94** | 98.45 | 48.06 | 91.11 | **27.93** | 95.15 |
| | LSUN-R | 15.77 | 97.39 | **6.67** | 98.42 | 82.23 | 77.38 | **67.36** | 86.59 |
| | iSUN | 12.66 | 97.70 | **4.16** | 98.85 | 80.23 | 79.01 | **53.33** | 90.10 |
| | Places | 21.56 | 96.52 | **6.31** | 98.70 | 86.03 | 75.32 | **51.14** | 88.03 |
| | DTD | 53.37 | 87.21 | **32.02** | 89.11 | 92.35 | 61.60 | **65.30** | 68.16 |
| | Tiny Imagenet | 27.43 | 95.85 | **11.81** | 97.86 | 76.37 | 80.50 | **58.25** | 89.13 |
| | SUN | 20.04 | 96.81 | **3.58** | 99.24 | 88.95 | 73.47 | **44.48** | 90.62 |
| | iNaturalist | 64.59 | 89.63 | **27.32** | 95.02 | 88.95 | 69.19 | **59.42** | 78.85 |
| | Average | 31.70 | 94.32 | **11.74** | 97.15 | 76.94 | 77.77 | **48.86** | 87.14 |
| ResNet34 | SVHN | 15.17 | 97.4 | **3.16** | 99.34 | 44.22 | 92.78 | **14.38** | 97.32 |
| | LSUN-C | 6.45 | 98.56 | **2.37** | 99.44 | 69.95 | 87.14 | **33.04** | 94.29 |
| | LSUN-R | 9.84 | 98.06 | **5.73** | 98.71 | 87.03 | 74.19 | **80.26** | 76.6 |
| | iSUN | 9.79 | 98.09 | **4.13** | 99.04 | 85.93 | 75.33 | **72.51** | 80.2 |
| | Places | 12.44 | 97.62 | **2.86** | 99.32 | 87.77 | 74.25 | **77.51** | 77.63 |
| | DTD | 25.46 | 94.82 | **12.76** | 96.57 | 87.69 | 70.78 | **68.59** | 72.49 |
| | Tiny Imagenet | 24.69 | 95.67 | **16.66** | 97.03 | 84.4 | 75.36 | **79.07** | 76.59 |
| | SUN | 10.65 | 97.97 | **1.47** | 99.63 | 91.56 | 70.84 | **77.94** | 77.88 |
| | iNaturalist | 10.79 | 97.74 | **5.05** | 99.08 | 88.22 | 80.24 | **66.22** | 80.29 |
| | Average | 13.92 | 97.33 | **6.02** | 98.68 | 80.75 | 77.88 | **63.28** | 81.48 |
| WRN40-2 | SVHN | 30.15 | 95.17 | **12.58** | 97.7 | 57.21 | 91.55 | **19.15** | 96.54 |
| | LSUN-C | **23.74** | 96.05 | 32.33 | 93.94 | 42.27 | 92.69 | **25.94** | 95.47 |
| | LSUN-R | 17.18 | 97.13 | **9.19** | 98.1 | 77.8 | 81.12 | **75.04** | 82.34 |
| | iSUN | 18.23 | 97.05 | **8.32** | 98.3 | 76.09 | 82.46 | **63.44** | 86.05 |
| | Places | 17.43 | 96.99 | **5.27** | 98.84 | 52.59 | 89 | **24.26** | 95.13 |
| | DTD | 26.25 | 95.11 | **13.26** | 96.1 | 51.65 | 86.94 | **25.17** | 93.27 |
| | Tiny Imagenet | 38.79 | 93.75 | **24.2** | 95.56 | 68.91 | 84.33 | **62.71** | 86.66 |
| | SUN | 16.29 | 97.23 | **3.51** | 99.18 | 55.59 | 88.68 | **20.76** | 95.95 |
| | iNaturalist | 30.46 | 95.69 | **5.05** | 98.84 | 21.81 | 96.3 | **9.02** | 98.11 |
| | Average | 24.28 | 96.02 | **12.63** | 97.40 | 55.99 | 88.12 | **36.16** | 92.17 |

# B   THEORETICAL ANALYSIS.

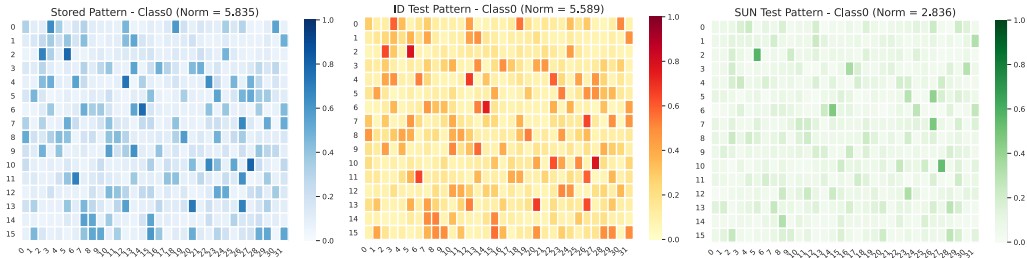

Figure 6: Heatmap of the Stored/ID/OOD pattern derived by ResNet18, the ID dataset, and OOD dataset is CIFAR10 and SUN (Xiao et al., 2010) respectively. For visualization, the pattern is re-shaped to 32*16 from the dimension of 512. From the figure, we can see the expectation of the ID/Stored pattern is larger than the expectation of the OOD pattern, which supports the theory mentioned below.

Let $\boldsymbol{y} := [y_1, y_2, ...y_m]$ and $\boldsymbol{z} := [z_1, z_2, ...z_n]$ be the output logits and the penultimate layer output, respectively. Here, $m$ is the dimension of logits output (i.e., the number of classes) and $n$ is the dimension of the penultimate layer output. We superscript the vector $id$ and $ood$ (e.g., $\boldsymbol{y}^{id}$ denotes logits output derived from an ID sample) to indicate the input type of the vector. For every $z_j^{ood}$ in $\boldsymbol{z}^{ood}$, we assume that they are independent random variables following the same Gaussian distribution, i.e., $z_j^{ood} \sim \mathcal{N}\left(0, \sigma^2\right)$. Let $[\boldsymbol{v}_1, \boldsymbol{v}_2, ...\boldsymbol{v}_m]$ be the weight matrix of the last layer (linear layer) with each $\boldsymbol{v}_j$ denotes the categorical vector for class $i$. Thus, we have $\boldsymbol{y} = [\boldsymbol{v}_1, \boldsymbol{v}_2..., \boldsymbol{v}_m]^T \boldsymbol{z}$.

For a test sample $\boldsymbol{\xi}$ from OOD data, we assume that it is classified as the category $k$, and we have the following formula. Among them, $\mathcal{M}$ represents a distribution of the maximum value of $m$ Gaussian random variables.

$$
\begin{aligned}
\boldsymbol{y}_k^{ood} &= \max\left(\boldsymbol{v}_1^T \boldsymbol{z}^{ood}, \boldsymbol{v}_2^T \boldsymbol{z}^{ood}, ...\boldsymbol{v}_m^T \boldsymbol{z}^{ood}\right) \\
\boldsymbol{y}_k^{ood} &\sim \mathcal{M}^{ood} = \max\left(\mathcal{N}\left(0, \sigma^2\|\boldsymbol{v}_1\|_2^2\right), \mathcal{N}\left(0, \sigma^2\|\boldsymbol{v}_2\|_2^2\right)..., \mathcal{N}\left(0, \sigma^2\|\boldsymbol{v}_m\|_2^2\right)\right) \\
\boldsymbol{y}_{q \neq k}^{ood} &\sim \mathcal{N}\left(0, \sigma^2\|\boldsymbol{v}_q\|_2^2\right)
\end{aligned}
\tag{9}
$$

For $\boldsymbol{y}_k^{id}$, we assume that it follows another distribution $\mathcal{I}$ whose expected value is a positive number larger than the expected value of $\boldsymbol{y}_{q \neq k}^{ood}$. When calculated by Eq. 6, the expectation calculated from output logits and penultimate layer outputs are shown as follows ($*$ denotes inner product):

$$
\begin{aligned}
\mathbb{E}\left[\boldsymbol{y}^{id} * \boldsymbol{y}^{ood}\right] &= \mathbb{E}\left[\boldsymbol{z}^{id^T} \boldsymbol{v}_k \boldsymbol{v}_k^T \boldsymbol{z}^{ood} + \sum_{j=1, j \neq k}^{m} \boldsymbol{z}^{id^T} \boldsymbol{v}_j \boldsymbol{v}_j^T \boldsymbol{z}^{ood}\right] \\
&= \mathbb{E}\left[\mathcal{I}^{id}\right] \mathbb{E}\left[\mathcal{M}^{ood}\right] + \sum_{j=1, j \neq k}^{m} \sum_{p=1}^{n} \sum_{q=1}^{n} \boldsymbol{v}_{jp} \boldsymbol{v}_{jq} \mathbb{E}\left[\boldsymbol{z}_p^{id}\right] \mathbb{E}\left[\boldsymbol{z}_q^{ood}\right] \\
&= \mathbb{E}\left[\mathcal{I}^{id}\right] \mathbb{E}\left[\mathcal{M}^{ood}\right] > 0
\end{aligned}
\tag{10}
$$

$$
\mathbb{E}\left[\boldsymbol{z}^{id} * \boldsymbol{z}^{ood}\right] = \mathbb{E}\left[\boldsymbol{z}^{id}\right] \mathbb{E}\left[\mathcal{N}^{ood}\right] = 0 < \mathbb{E}\left[\boldsymbol{y}^{id} * \boldsymbol{y}^{ood}\right]
\tag{11}
$$

Therefore, compared with using the penultimate layer as the pattern to calculate the energy function, logits output will assign OOD samples with higher scores. Therefore, it will be more challenging to distinguish ID samples from OOD samples.

# C   ABLATION EXPERIMENT.

The ablation experiment consists of two parts. First, we evaluate different metrics instead of the inner product which is used in our approach to measuring the similarity of the stored pattern and test pattern. To be specific, we use "Euclidean Distance" and "Cosine Similarity", the results are shown in Table. 12. Second, we use the output from other layers (e.g., layer3 in ResNet) instead of the penultimate layer (layer4 in ResNet) to act as the representation of our approach. The results are shown in Table. 13.

Table 12: OOD detection performance with different metrics, the ID dataset is CIFAR10. From the results, the inner product used in our approach performs better than another two metrics.

| Backbone Network | OOD Datasets | Metric | | | | | |
|---|---|---|---|---|---|---|---|
| | | Euclidean Distance | | Cosine Similarity | | Inner Product (Ours) | |
| | | FPR95 (↓) | AUROC (↑) | FPR95 (↓) | AUROC (↑) | FPR95 (↓) | AUROC (↑) |
| ResNet18 | SVHN | 38.59 | 93.76 | 39.59 | 93.70 | **5.87** | 98.74 |
| | LSUN-C | 22.57 | 96.07 | 13.22 | 97.65 | **7.94** | 98.45 |
| | LSUN-R | 20.50 | 96.35 | 17.95 | 96.93 | **6.67** | 98.42 |
| | iSUN | 14.72 | 97.14 | 14.90 | 97.35 | **4.16** | 98.85 |
| | Places | 12.18 | 97.53 | 17.73 | 96.84 | **6.31** | 98.70 |
| | DTD | 13.59 | 97.22 | **13.42** | 97.57 | 32.02 | 89.11 |
| | Tiny Imagenet | 27.65 | 95.01 | 25.15 | 95.53 | **11.81** | 97.86 |
| | SUN | 8.98 | 98.12 | 12.89 | 97.68 | **3.58** | 99.24 |
| | iNaturalist | **21.69** | 95.84 | 22.04 | 96.19 | 27.32 | 95.02 |
| | Average | 20.05 | 96.34 | 19.65 | 96.60 | **11.74** | 97.15 |
| ResNet34 | SVHN | 15.78 | 97.03 | 28.70 | 95.52 | **3.16** | 99.34 |
| | LSUN-C | 11.78 | 97.71 | 13.94 | 97.62 | **2.37** | 99.44 |
| | LSUN-R | 16.31 | 96.86 | 22.79 | 96.00 | **5.73** | 98.71 |
| | iSUN | 14.39 | 97.15 | 23.16 | 95.95 | **4.13** | 99.04 |
| | Places | 19.09 | 96.26 | 20.69 | 96.27 | **2.86** | 99.32 |
| | DTD | **10.28** | 98.01 | 12.07 | 98.00 | 12.76 | 96.57 |
| | Tiny Imagenet | 30.16 | 94.04 | 37.07 | 92.76 | **16.66** | 97.03 |
| | SUN | 14.92 | 96.92 | 16.80 | 96.81 | **1.47** | 99.63 |
| | iNaturalist | 8.11 | 98.37 | 9.36 | 98.40 | **5.05** | 99.08 |
| | Average | 15.65 | 96.93 | 20.51 | 96.37 | **6.02** | 98.68 |
| WRN40-2 | SVHN | 42.96 | 92.27 | 39.79 | 93.21 | **12.58** | 97.70 |
| | LSUN-C | 42.92 | 91.67 | 38.33 | 93.11 | **32.33** | 93.94 |
| | LSUN-R | 31.19 | 94.57 | 29.55 | 94.77 | **9.19** | 98.10 |
| | iSUN | 29.35 | 94.96 | 28.51 | 95.01 | **8.32** | 98.30 |
| | Places | 35.18 | 93.55 | 38.38 | 93.15 | **5.27** | 98.84 |
| | DTD | 14.41 | 97.25 | 15.74 | 97.27 | **13.26** | 96.10 |
| | Tiny Imagenet | 48.18 | 90.67 | 43.30 | 91.78 | **24.20** | 95.56 |
| | SUN | 30.93 | 94.43 | 35.22 | 93.87 | **3.51** | 99.18 |
| | iNaturalist | 23.50 | 95.77 | 20.35 | 96.53 | **5.05** | 98.84 |
| | Average | 33.18 | 93.90 | 32.13 | 94.30 | **12.63** | 97.40 |

Table 13: OOD detection performance comparison of shallow layer (the layer before penultimate layer) and penultimate layer as the representation. All values are averaged over the nine OOD datasets.

| Backbone Network | CIFAR10 | | | | CIFAR100 | | | |
|---|---|---|---|---|---|---|---|---|
| | shallow layer | | penultimate layer | | shallow layer | | penultimate layer | |
| | FPR95 (↓) | AUROC (↑) | FPR95 (↓) | AUROC (↑) | FPR95 (↓) | AUROC (↑) | FPR95 (↓) | AUROC (↑) |
| ResNet18 | 86.83 | 34.25 | **11.74** | 97.15 | 86.75 | 43.43 | **48.86** | 87.14 |
| ResNet34 | 83.87 | 51.30 | **6.02** | 98.68 | 73.77 | 60.83 | **63.28** | 81.48 |
| WRN40-2 | 91.84 | 47.13 | **12.63** | 97.40 | 89.97 | 48.65 | **36.16** | 92.17 |

# D    EXPERIMENTAL DETAILS.

**Software and Hardware.** The experiments are performed on one linux server (Operation system: Ubuntu Linux 18.04.1). For GPU resource, four NVIDIA GeForce RTX 3090 are used for ResNet and WRN. The python environment is 3.7 and the libraries we use to implement our experiments is PyTorch 1.12.1 ,

**Number of Evaluation RunsReported** Following (Liu et al., 2020; Hendrycks et al., 2018), performance for each OOD dataset is averaged over 10 random batches of samples.

**Training Details.** During the training procedure, some data augmentation (e.g., rotation, flipping, resizing) and training techniques (e.g., learning rate decay) are adopted to improve the model's accuracy. In more detail, we set the batch size as 128, the image size as 112 for ResNet, and 64 for WRN, respectively. And we use SGD as the optimizer with 0.1 as the initial learning rate and apply the learning rate decay at epochs 50 and 75, respectively.

