# OpenReview forum: "Out-of-Distribution Detection based on In-Distribution Data Patterns Memorization with Modern Hopfield Energy"
_ICLR.cc/2023/Conference — ICLR 2023 poster_

### Official Review · Reviewer_ANA9 · 2022-10-13

**Confidence:** 3
**Correctness:** 3
**Technical Novelty And Significance:** 3
**Empirical Novelty And Significance:** 3
**Recommendation:** 6

**Clarity, Quality, Novelty And Reproducibility:**

I think the presentation of this paper is clear, the proposed method is novel, and enough details have been given to reproduce the method.

**Strength And Weaknesses:**

Strength:
1. The paper is well-written and easy to follow.
2. The experiment result is quite significant and impressive, demonstrating the strong performance of the proposed method.
3. Further analysis of the penultimate layer versus the logits layer is also interesting.

Weaknesses:
1. The authors mention that "the energy function of Hopfield Network is well suited as a desirable measure of the discrepancy between the OOD sample and the stored patterns". Could the authors elaborate more on the reason behind this statement? Specifically, there exists a lot of metrics that can measure the discrepancy between features, such as measuring the distance of mean, the distance of covariance, the earth mover's distance, and so on. Thus, I am curious on is there a specific reason that the authors choose to use the energy function of the Hopfield Network to measure the discrepancy.
2. To further elaborate on my point 1, I think the idea proposed in this paper is quite similar to the metric learning methods in other areas (e.g., few-shot learning). Could the authors also discuss the relationship between the proposed method and metric learning?
3. Another small confusion I have is about the title. Does the "efficient" here imply that SHE is more efficient than HE or SHE is more efficient than the other methods as well? If the formal case, I suggest the authors delete the word "efficient" in the title as it is slightly misleading, if the latter case, I hope that some ablation study w.r.t. efficiency can be given to demonstrate this.

**Summary Of The Paper:**

This paper proposes a store-then-compare paradigm for out-of-distribution detection. The proposed method is motivated by Modern Hopfield Energy, and a simplified version of the proposed method is also derived.

**Summary Of The Review:**

Overall, while I have concerns about whether the proposed method is just an alternative to metric learning, I think even if that is the case, the proposed method is still interesting in the field of OOD detection. Thus, I tend to accept this paper.

---

> ### Comment · Reviewer_ANA9 · 2022-11-24
> **Thanks the authors' responses**
>
> Thanks to the authors for their further explanation. After reading the responses from the authors, I think they largely address my concern and I decide to keep my positive rating.

---

### Official Review · Reviewer_bCQs · 2022-10-22

**Confidence:** 3
**Correctness:** 3
**Technical Novelty And Significance:** 2
**Empirical Novelty And Significance:** 2
**Recommendation:** 6

**Clarity, Quality, Novelty And Reproducibility:**

Overall, the paper is clearly written and is easy to follow. To me, the motivation is a little bit of unclear and the novelty is limited. From my opinion, addressing these issues can be helpful in improving the quality of the paper. Besides, I did not check the reproducibility of the paper.

**Strength And Weaknesses:**

> Strength:

To the best of my knowledge, this paper is the pioneer in using the modern Hopfield networks in OOD detection. The authors interpret the energy function as the measure of similarity, which can be used to discern ID and OOD data. Further, the authors suggest a simplified realization named SHE, which approximate the original energy function by the Taylor series, which eases the burden in memorizing all embedding features in energy calculation.

> Weakness:

- *The motivation of the proposed method is not well supported*. The authors claim the deficiencies of previous works in using logit outputs. However, the discussion is only valid for their proposed method in using the energy function. To fully demonstrate the deficiencies in using logit outputs, I think the authors could take other advanced works as examples, e.g., [1,2]. To me, these methods can also demonstrate promising results regarding various OOD setting. Further, I think there are also many other works that use the embedding features in OOD detection [3], which may challenge the novelty of the paper.

[1] Yiyou Sun, et al. Out-of-distribution Detection with Deep Nearest Neighbors. ICML'22.

[2] Vikash Sehway, et al. SSD: A Unified Framework for Self-supervised Outlier Detection. ICLR'21.

[3] Kimin Lee, et al. A Simple Unified Framework for Detecting Out-of-distribution Samples and Adversarial Attacks. NeurIPS'18.

- *The novelty of the proposed method is limited*. Both the energy functions [4] and the so-called store-then-compare paradigms [5] have been studied. Therefore, I think it can improve the quality of the paper in discussing the superiority of the proposed method over these representative baselines.

[4] Weitang Liu, et al. Energy-based Out-of-distribution Detection. NeurIPS'20.

[5] Yiyou Sun, et al. Out-of-distribution Detection with Deep Nearest Neighbors. ICML'22.

- *The experiments are not enough*. I do not think the authors conduct extensive experiments that can fully support the effectiveness of the proposed method. Experiments with large semantic space [6] (e.g., ImageNet benchmark) and hard OOD scenario [5] (e.g., CIFAR-10 vs. CIFAR-100) are not considered. Further, comparison with advanced baseline methods (e.g., [1,2,5]) is also missing.

[6] Rui Huang and Yixuan Li. MOS: Towards Scaling Out-of-distribution Detection for Large Semantic Space. CVPR'21.

**Summary Of The Paper:**

This paper studies out-of-distribution detection, aiming at making classification models excel at discerning ID and OOD data. It is an important problem for safety-critical applications, and has attracted increasing attention recently. In this paper, the authors state that previous OOD methods estimate the OOD confidence in the logit space with tedious hyperparameters, which are hard to be deployed in reality. To this end, the authors suggest a new store-then-compare paradigm. It is motivated by the modern Hopfield networks for the discrepancy score calculation, which is free from hyper-parameters and is easy in deployment. The authors conduct experiments on nine widely-used OOD datasets, and the authors claim their superiority over the state-of-the-arts.

**Summary Of The Review:**

This paper adopts the Hopfield network in OOD detection, which may have some novelties to benefit the community. However, to improve the quality of the paper, it will be great if the authors could address my concerns in the Summary of the Paper.

---

> ### Comment · Reviewer_bCQs · 2022-11-22
> **Thanks for the authors' response**
>
> The authors have largely address my previous concerns. So, I would like to raise my score to 6.

---

### Official Review · Reviewer_boCN · 2022-10-25

**Confidence:** 4
**Correctness:** 3
**Technical Novelty And Significance:** 3
**Empirical Novelty And Significance:** 3
**Recommendation:** 6

**Clarity, Quality, Novelty And Reproducibility:**

The paper is written clearly, the proposal is novel, numerical experiments look convincing.

**Strength And Weaknesses:**

The premise of Hopfield networks is to place plausible data points at the local minima of the energy function. The authors propose to use this property for detecting datapoint that are located far away from those local minima, which correspond to in-distirbution training examples. Overall, this is a very natural idea, and the empirical evaluation supports its usefulness. I have a few suggestions that I describe below that may improve the proposed method and the quality of the presentation.

Throughout the paper the authors focus on one specific model from the modern HN family. This model uses the dot-product similarity function between the memories $S$ and the datapoints $\xi$.  Although this is a plausible measure, it is unclear why the representations in the penultimate layer of the front-end networks (e.g. ResNet18) would respect this measure. I suspect that a much more natural measure of the similarity would be the Euclidean distance between the memories and the data points, see for example [(Millidge et al., 2022)](https://arxiv.org/abs/2202.04557) for a description of the HN with this similarity measure. A third possible choice would be a cosine similarity between the memories and the new test datapoints. I suspect that among these three networks the one with the Euclidean distance will demonstrate the best performance in terms of out-of-distribution sample detection, the dot-product based network (considered in this work) would be second, and the cos-similarity would perform the worst among these three. I would like to see the results of these ablation studies in the revised manuscript.

The description of the prior work on Hopfield networks is not entirely accurate. Binary HN were introduced in [(Hopfield PNAS 1982)](https://www.pnas.org/doi/10.1073/pnas.79.8.2554), continuous HN were introduced in [(Hopfield PNAS 1984)](https://www.pnas.org/doi/10.1073/pnas.81.10.3088). Modern HN (both continuous and binary) were introduced in (Krotov & Hopfield 2016). The contribution of (Ramsauer et al., 2020) was to generalize the theory of (Krotov & Hopfield 2016) to the case of softmax activation function, and point out the relationship with transformers. For example, the first sentence of the second paragraph of section 3.2 is misleading - HN have been used for continuous pattern retrieval since 1984, see [(Hopfield PNAS 1984)](https://www.pnas.org/doi/10.1073/pnas.81.10.3088).


**Summary Of The Paper:**

The paper proposes to use the energy function of modern Hopfield networks to detect out-of-distribution datapoints.

**Summary Of The Review:**

Nice paper, I would like to see the results of the ablation studies that I requested, and revisions regarding the more accurate description of various models of the Hopfield family.

---

### Official Review · Reviewer_sLwk · 2022-11-01

**Confidence:** 4
**Correctness:** 3
**Technical Novelty And Significance:** 3
**Empirical Novelty And Significance:** 3
**Recommendation:** 6

**Clarity, Quality, Novelty And Reproducibility:**

Clarity: good
Quality: fair
Novelty: good
Reproducibility: good


**Strength And Weaknesses:**

strengths
* The algorithm is intuitive and simple with strong empirical results, writing is easy to follow.

weakness
* Lack of analysis and understanding on when the algorithm should be better than baseline, there are huge tables on many datasets and backbone networks, but not much insights on more granular analysis.
* Lack empirical analysis and visualization on the stored patterns
* no results with ImageNet dataset with 1k+ classes, which seems a critical aspect of the algorithm: number of classes, especially given that the performance seems not as good on CIFAR100 than on CIFAR10.



More details below:

I don’t feel the need to motivate the algorithm with Hopfield network. At the end of the day, it seems all it does is to store the average outputs from the penultimate layer for each class, and during test time, we compute the inner product between the test blob and the stored “pattern” for the predicted class. I don’t feel motivating it with Hopfield energy is necessary, especially as there is a series of approximations not fully justified with empirical data demonstrated.

What I would prefer is to just start with the algorithm and briefly mention it could be motivated from Hopfield energy and put section 3.2-3.4 in the appendix, and leave more space for empirical analysis and visualization of the patterns remembered? For example, could we calculate the distances between classes and visualize the patterns in 2-D (or 3-D), and see where the ID and OOD examples land? even more interestingly, how are they evolving during training?

It is misleading to say “Similar performance gain can be observed when CIFAR100 is used as the ID training data” – if we look at Table 5 in Appendix A, where with ResNet34, ReAct is around 6% better on average. This raises several questions we should study, e.g., why on certain training sets the OOD performance is better,  why on certain backbone networks the OOD performance is worse than baseline, etc.. I believe these are important analyses to be done. Also interesting to see results on ImageNet with even more classes – Will the algorithm still perform better? or is it only good in the small-num-classes regime?

The proposed algorithm is not w/o any hyperparameters: we still need a threshold deciding what are OOD examples? OOD examples can be of any form. How do we determine such a threshold? is the threshold the same for each class or different for different classes?

About why the penultimate layer is better than the logit layer, I find the intuitive argument more convincing than the theoretical analysis. First, there are two strong assumptions w/o any justification: (1) z follows zero-mean Gaussian distribution, (2) $y_k^{id}$ follows another distribution $\cal{I}$ whose expected value is a positive number larger than the expected value of $y_{q\not=k}^{ood}$. At least, show empirical histograms to justify such assumptions? Second, there seems not much point to compare the inner products of two layers, what we care about is the relative values of the inner product of ID vs OOD examples in the same layer.

typos
* “Intuitive Explanation.Given”, add space
* section 4.3.4, “computation(Hendr”, add space



**Summary Of The Paper:**

This paper proposes a novel out-of-distribution (OOD) detection algorithm with “store-and-compare” fashion: store the average pattern in the penultimate layer for each class, and during test, compute the inner product for the test blob with the stored pattern for the predicted class, and this inner product will be thresholded to determine OOD or not. The authors try to motivate the algorithm from Hopfield energy with a series of approximations, and also argue that the penultimate layer is more effective than the logit layer as used in most existing work. Experiments are conducted on vision datasets (two training sets CIFAR10 and CIFAR100, and 9 OOD sets), with comparison with three baselines, and demonstrated superior results most of the time.

**Summary Of The Review:**

This paper proposed a novel OOD algorithm with an intuitive idea that simply compares the test examples with stored average patterns for the predicted class on the 2nd-to-last layer. It attempts to motivate it from Hopfield energy, which I don’t think is necessary; the intuitive argument that the penultimate layer is better than the logit layer is convincing and well supported by experiments, but theoretical analysis seems futile. Experiments are relatively comprehensive in terms of datasets, baselines and average performance, but not results with ImageNet dataset as state-of-the-art method ReAct did, and lack insights on when the algorithm should perform better than baseline. The writing is clear and easy to follow, w/o little typos. Overall I can see this becoming a good paper, but not in current form.

---

### Decision · Program_Chairs · 2023-01-20

**Decision:**

Accept: poster

**Justification For Why Not Higher Score:**

See meta review

**Justification For Why Not Lower Score:**

NA

**Metareview: Summary, Strengths And Weaknesses:**

This paper investigates an important problem of out-of-distribution (OOD) detection, which has attracted increasing research attention for the safe deployment of machine learning models. The novel approach here is to introduce the energy function from a Hopfield Network for OOD detection. This energy function is defined based on the _log-sum-exp_ of inner products between the test feature and the training features of the predicted class (referred to as ``stored patterns'' by the authors).

Based on the review comments and my own take on the paper, I will summarize the strengths and weaknesses below in detail:

**Strengths**

1. The algorithm is simple to implement and achieves strong empirical results, especially on the CIFAR-10 dataset.
2. The paper presents extensive ablations and comparisons that help further understand the advantage of the proposed algorithm. Recent SOTA post-doc methods such as ReAct [1] are also compared, which is informative.
3. The algorithm is evaluated on a broad spectrum of 9 OOD test datasets, which is comprehensive.
4. The writing and presentation are very clear overall. It's pleasant to read.
5. Authors actively engaged in the discussion and made effort to incorporate further suggestions made by the reviewers. Some newly added experiments and discussions have strengthened the manuscript, including evaluation on ImageNet, and ablations on different distance metrics.

**Weaknesses and suggestions for final revision**
1. As reviewer sLwk has brought up, the authors left out the comparison with ReAct on ImageNet-1k for no good reason. This creates inconsistency in experiments, considering ReAct was compared for both CIFAR-10 and CIFAR-100. Referencing Table 1 in [1], the average FPR95 appears to be 31.43%, which appears to be stronger than the proposed SHE score (44.71%). This is all fine. I don't think the paper should be rejected because of this. With that being said, I highly encourage the authors to be transparent about this, and accordingly, tune down the claim of SOTA for ImageNet-1k (and perhaps CIFAR-100 as well, where the results also seem mixed).
2. Related to the first comment, Reviewer sLwk also questioned whether the proposed method is more effective in small-class regimes. The question was not directly answered in the rebuttal. The current evaluations on CIFAR-100 and ImageNet-1k do display a more limited efficacy. Hence, it would be valuable for the community if the authors can provide an explicit limitation section, discussing where the method is better suited.
3. Echoing reviewer ANA9's suggestion, the wording "efficiency" in the title & main body does not capture the core contribution and property of this work, and should be removed. From a post hoc perspective, all baselines including MSP, energy score, and ReAct share a similar inference complexity and time. Both MSP and energy score are also hyper-parameter free. Hence, the "efficiency" claim doesn't seem compelling or well-supported, unless some ablation study w.r.t. efficiency can be given to demonstrate this.
4. It would be helpful to have a clear discussion section on the conceptual relation to [2], given it's the closest work. It seems both works define the energy score using the _log-sum-exp_ operation, yet the crucial differences lie in using logits (summed across classes) vs. inner products (sum across sample populations, for predicted class). Discussing the two methods together would facilitate understanding of the proposed method.
5. The Taylor approximation in Section 3.4 is not rigorous. Please note that the expansion $\log (1+x)=x- ((x^2)/2)+((x^3)/3)-((x^4)/4)+..$ holds if $|x| <1$. Whether this condition holds is unclear. Your final step in Equation (5) is almost unnecessary. It loses the mathematical meaning while giving similar computation costs. Why not use the $\log (1+ \xi^\top \cdot \frac{\sum e_{ij}}{N_j})$?

All four reviewers are supportive of this work. I think overall it's an interesting and solid contribution to the research community. While there are some lingering concerns, the overall merits outweigh them. Given most of the concerns are fixable, I recommend acceptance conditioned on the above changes made.

Congratulations!

[1] Sun et al. ReAct: Out-of-distribution Detection With Rectified Activations. NeurIPS 2021.

[2] Liu, et al. Energy-based Out-of-distribution Detection. NeurIPS 2020.







**Note From Pc:**

if the above contains the word "oral" or "spotlight" please see: "oral" presentation means -> notable-top-5% and "spotlight" means -> notable-top-25%. As stated in our emails, we are disassociating presentation type from AC recommendations